# Acting Instead of Reacting—Ensuring Employee Retention during Successful Introduction of i4.0

**Steffen C. Eickemeyer** [1], **Jan Busch** [2], **Chia-Te Liu** [1] **and Sonia Lippke** [3,*]

[1] Department of Business Studies, Jacobs University, 28759 Bremen, Germany; s.eickemeyer@jacobs-university.de (S.C.E.); chandler20708@gmail.com (C.-T.L.)
[2] Mechanical Engineering at Miele & Cie. KG, 31275 Lehrte, Germany; jan.busch87@gmx.de
[3] Department of Psychology and Methods, Jacobs University, 28759 Bremen, Germany
[*] Correspondence: s.lippke@jacobs-university.de; Tel.: +49-421-200-4730

**Abstract:** The increasing implementation of digital technologies has various positive impacts on companies. However, many companies often rush into such an implementation of technological trends without sufficient preparation and pay insufficient attention to the human factors involved in digitization. This phenomenon can be exacerbated when these technologies become highly dependent, as during the COVID-19 pandemic. This study aims to better understand challenges and to propose solutions for a successful implementation of digitized technology. A literature review is combined with survey results and specific consulting strategies. Data from the first wave of the COVID-19 pandemic in Germany were collected by means of an online survey, with a representative sample of the German population. However, we did not reveal any correlation between home office and suffering, mental health, and physical health (indicators of digitization usage to cope with COVID-19 pandemic), but rather that younger workers are more prone to using digitized technology. Based on previous findings that older individuals tend to have negative attitudes toward digital transformation, appropriate countermeasures are needed to help them become more tech-savvy. Accordingly, a software tool is proposed. The tool can help the management team to manage digitization efficiently. Employee well-being can be increased as companies are made aware of necessary measures such as training for individuals and groups at an early stage.

**Keywords:** change management; COVID-19; decision-support model; digitization; employee motivation; employee satisfaction; human resources; Industry 4.0; software tool

## 1. Introduction

In the period of the COVID-19 pandemic, the dependency on digital technologies became heavily apparent, particularly due to the physical distancing policies imposed [1,2], with effects on operations and production management. Overall, digitization in general is progressing steadily and COVID-19 illustrates this clearly. Even traditional entrepreneurs, who have not created options for mobile/home office or similar before COVID-19, have largely changed their views in recent months. However, the importance of digitization in operations and production management was evident long before. Especially due to the high cost pressure (to increase the degree of value creation in business processes), the increased complexity in products and processes, and the demand for higher efficiency in satisfying customer requirements, the use of Industry 4.0 (i4.0) technologies is becoming increasingly important. As i4.0 technologies are amongst others created to deal with relatively complex processes, for example, creating small and highly customized batch sizes, it could enable companies to gain very profitable and competitive advantages. While i4.0 offers success, companies are faced with some challenges when implementing it [3].

One of the many challenges that companies face is the emergence of human symptoms induced by a new and innovative work environment. Under the impact of i4.0, working individuals must confront the autonomy of actions and excessive demands, as well as the

adaptation to new responsibilities and roles inside the new processes [4] (Gorecky et al., 2014). This can result in mental stress [5–7], demotivation, and conflicts with superiors, colleagues, and one's team [8,9]. The actual causes of the symptoms are not perceived by companies in depth [10,11]. According to a systematic literature study by Kadir, Broberg, and Conceição, reviewing over 80 studies, there is a lack of attention to human factors in the context of i4.0 [12]. In general, little systematic scientific research is available on the influence of digitization on employees' adverse experiences [12]. In addition, many companies have poor knowledge of i4.0 technologies, which might still increase companies' inappropriate dealings with employees' uncertainties. For instance, a survey study about i4.0 in the manufacturing sector of New Zealand in 2018 shows that 33% of the manufacturers have a "poor" or "very poor" knowledge of i4.0 [13]. Uncertainties and unawareness certainly do not assist employees in the integration of change management. As a result, the impacts of human symptoms and the causes of the symptoms are the key drivers of this fold of the challenge. Accordingly, Hilb expressed his views on digitization and artificial intelligence (AI): "While the business appetite for AI is clear and the advances in technology are certain, it will ultimately be the social dialogue that will be critical. In this respect, companies will have to prove that they are aware of their responsibility in dealing with AI in order to win the trust of society" [14] (p. 867). Based on this, Hilb gives the following management perspective: "Today's boards of directors can play a central role in this process if they are willing and able to take the driver's seat" [14] (p. 867).

The aim of this paper is therefore (a) to review how it is possible to recognize in advance that digitization could overburden employees and therefore might lead to lower productivity. In addition, we aim to (b) investigate how these negative developments can be counteracted, especially in times of crisis, i.e., when the COVID-19 pandemic urged many organizations and individuals to shift to mobile work. Therefore, this paper will (a) review the literature and (b) present empirical data on individuals' perceptions of mobile work and psychological reactions to it, and based on this, (c) explain a self-developed decision-support model with a software tool for top management.

The article is structured in 6 sections: the following, Section 2, presents a literature review to classify the article thematically. Section 3 describes the empirical evidence with the materials and methods used. Based on this, the results of the survey are presented in Section 4. The findings and the relationships described are discussed in Section 5 and used to develop a decision-support model for dealing with employees during digital transformation. Finally, Section 6 contains the conclusions, which provide both a summary of the article and an outlook on subsequent research work.

## 2. Literature Review

The following is a summary of the areas of literature that are essential to this paper. First, an understanding of digitization (Section 2.1), which is central to this article, is created. In the context of this, the central buzzword, Industry 4.0, which is the driver for many industry initiatives, is brought into focus. Section 2.2 provides a general explanation of human behavior during change processes. These two sections help to provide a basic understanding, so that Section 2.3 can then focus specifically on effects on employees during the implementation of Industry 4.0. These effects on employees and the optimal way to deal with them are the main topic of this article.

### 2.1. Industry 4.0

The term "Industry 4.0" (originally "Industrie 4.0") was first coined publicly by the German government in the year 2011 [15]. The term described neither existing developments nor current research. Together with representatives from business, politics, and academia, the government initiated the concept of i4.0 to enhance the competitiveness and innovation of Germany's domestic manufacturing industry [16]. The concept of i4.0 as the fourth industrial revolution, was, in fact, predicted ex-ante rather than recognized ex-post [17]. This is unique because the other three great industrial revolutions were always

named, described, and analyzed in the aftermath. "Even though Industry 4.0 is one of the most frequently discussed topics these days, I could not explain to my son what it really means", said Audi's production site manager [16]. As a consequence, many studies still claim that there is no generally accepted definition for i4.0 [16,18,19].

In order to ensure a common understanding, we will share short definitions (according to [20]) regarding the core drivers of the four industrial revolutions. The first revolution (1784) was based on water energy and steam engines to power machines. The second revolution (1870) used electrical energy to enable mass production by continuously driven conveyor systems/belts. The third revolution (1969) used programmable logic controllers (PLC) and IT systems for the automation of complete value-added processes. The fourth revolution (today) uses these four main levers: Internet of Things (IoT), Industrial Internet of Things (IIoT), Cloud-based Manufacturing (CbM), as well as Smart Manufacturing (SM) [20,21]. While i4.0 does not revolve around a single innovation that enables the shift, the four main levers comprise various numbers and categories of technologies, such as Big Data analytics, Artificial Intelligence (AI), Cyber-Physical Systems (CPS), and decentralized systems [1,19]. The i4.0 paradigm promotes the connection of sensors, devices, and enterprise assets, both to each other and to the internet [22].

The main difference between i4.0 and Computer-Integrated Manufacturing (CIM) is the concern of the human role in the production environment [20]. In i4.0, human workers play an important role in carrying out production, while CIM considers production without workers [20,23]. This underlines the central role of human resources in value-creation processes in the era of i4.0. Accordingly, these human performance factors should be considered appropriately in change management.

## 2.2. Human Behavior during Change Processes

In the 2015 research report, "Skills for Digital Transformation", more than 84% of the interviewed individuals shared a strong consensus on business change management skills being of major importance for companies' transformations [24]. The introduction of i4.0 tools usually requires changes in the work processes of employees. Increasing complexity in processes due to i4.0 technologies could have a negative impact on employees (e.g., blurred work–non-work boundaries, higher workload, new responsibilities, fear of job loss, etc.). Accordingly, Cinquini commented on i4.0, AI, and Big Data: "Artificial Intelligence (AI) and Big Data might indicate how decision-making would be uncontextualized, and people and local setting may become irrelevant. Thus, the questions about the consequences for human actors and their roles in pervasive digitalization and how can we develop fair and valid performance management instruments for making managers and employees accountable are important." [25] (p. 847). Respectively, companies need to cope with these concerns to maintain high employee productivity, and with that actually make use of the potentials of i4.0 by means of employees' participation in dynamic operational performance [26].

There are different models for human behavior during change processes. The following discussion utilizes the Kübler-Ross "Change Curve" [27]. This curve (Figure 1) is based on a model originally developed in the 1960s by Elisabeth Kübler-Ross, and is fully in line with the principle to have employees participate in continuous improvement to ensure operational performance [26]. The seven steps of the change curve describe the usual reactions of employees to changes in their work processes.

However, this knowledge of the exact objectives of change measures in the field of digitization is an absolute prerequisite for good integration of employees in the change process. Only if the goal is clear and the possible effects can be modeled, can the path to the goal also be described, which will make it possible to integrate and retain employees (vs. demotivation, termination, or similar). In order to achieve this, the goals and tools for the introduction of i4.0 methods as well as their effects on the employees must be considered in more detail.

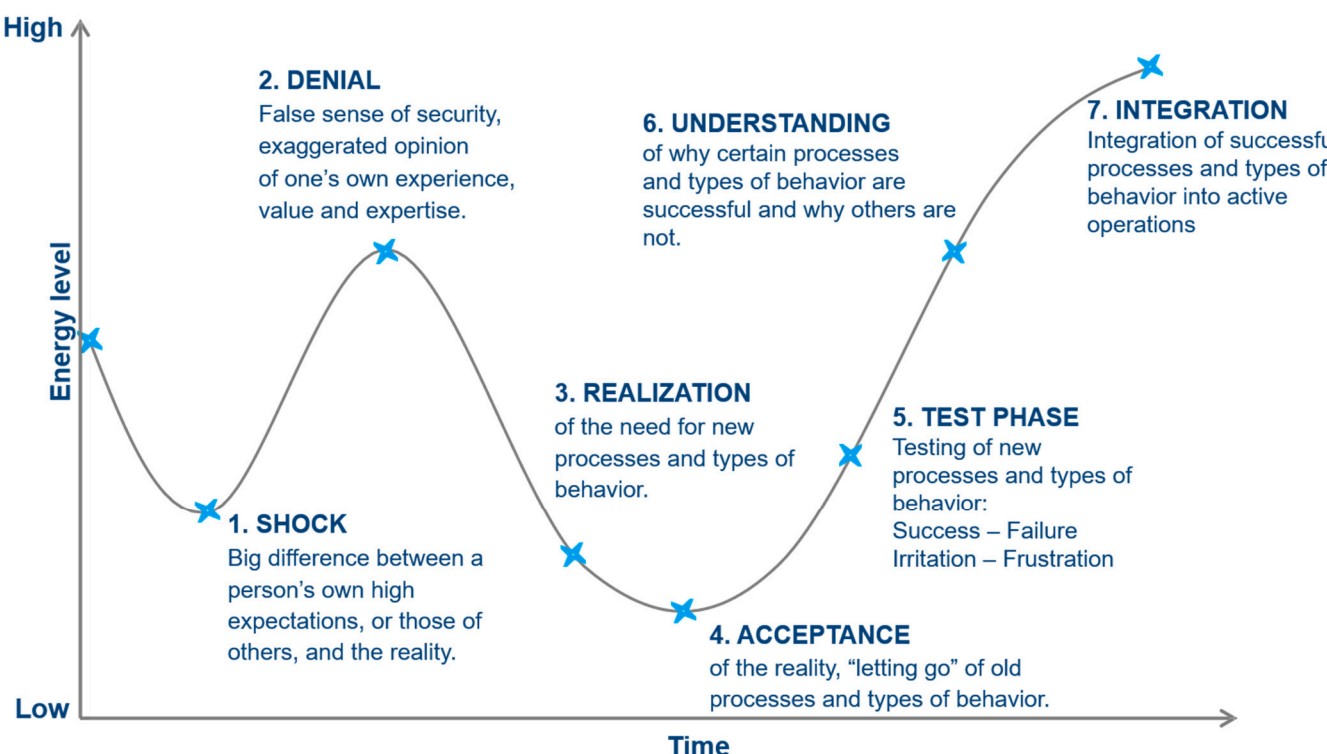

**Figure 1.** Decision-support model for dealing with employees during digital transformation (based on [27]).

### 2.3. Effects on Employees during Implementation of Industry 4.0

As described at the beginning, i4.0 is not defined by mutual agreement, nor do complete real value chains exist that fully comply with the usual i4.0 standards. Nevertheless, it is important to understand what experiences of the impact of the implementation of i4.0 on employees have already been described and analyzed. Some studies established maturity models or company-readiness guidelines for i4.0. The process model in small and medium enterprises (SMEs), as a guiding framework to implement i4.0 proposed by Ganzarain and Errasti suggests, is a three-stage maturity model: "Envision", "Enable", "Enact". In summary, it is dedicated to, firstly, defining the understanding of i4.0 in the company's capacity and customers' needs, and secondly, forming a roadmap that enables strategy formulation, which then transforms into the company's actual business model [28].

Another approach by Pessl et al. contains a six-step roadmap as a holistic approach to implementing i4.0. Focusing on the human, it suggests the following procedure: start-workshop, analyze i4.0 maturity, define the target state, define and evaluate measures, prepare and realize decisions, and define projects [29]. Most studies focus on the impact and readiness aspects of the company at the organizational level. In terms of technological transformation, words such as "digitization anxiety" and "technophobia" show up consistently during the research of relevant topics. Researchers have defined the symptoms in broader terms as computer anxiety and technophobia [30,31], especially for older employees.

As rather little systematic research can be found so far on the impact of i4.0 enabling employees to work from home and options for preventing its negative impact, this study aims to better understand how the transition to a remote work environment in Germany due to COVID-19 affected employee experience: Are there differences between employees working from home and employees working on-site regarding age, gender, and children, as well as their perceptions of feeling negatively affected by the COVID-19 pandemic and their well-being/health in general?

## 3. Materials and Methods

To test, a secondary data analysis was performed with the study described in detail elsewhere [32]. However, the aspects relating to working from home were not analyzed before. The survey, by means of online questionnaire, was launched in June 2020, to collect data from a sample of individuals aged 18 years and above with a sample size of N = 1050, by Bilendi GmbH on behalf of Weleda. A representative sample of inhabitants in Germany was recruited. The questionnaire-based survey was conducted between 8 June and 15 June in the year 2020. Using email, the company contacted individuals in their database by means of stratified sampling. In the inviting email, the purpose of the study was explained. When individuals clicked on the link to the study, they received further participant information which included a confidentiality statement. After reading this, study participants were asked to give informed consent by filling in a form. Only those who provided informed consent were allowed to continue to the questionnaire. The reason for this methodology and advantages of its choice were that the research questions could be investigated at high quality and there was easy access to the sample even in times of restrictions due to the COVID-19 pandemic. The main aim was to be able to generalize the findings to the larger population and to overcome bias in typical respondent behaviors.

Different subject areas were collected, such as socio-demographic data, work situation, perceived stress in times of lockdown, and perceived changes in the "new" everyday life. For the wording of the analyzed questions, see Tables 1 and 2 The way of designing the questionnaire, in order to generate the necessary data to achieve the research objectives, was determined by a theory-based and co-creative process: the last author performed a scoping review of the literature and, in exchange with potential future study participants, items and response formats were piloted and partially adapted. The validation of the questionnaire was performed by means of face validity and pilot analyses due to limited options to cross-validate the measures with other items.

Overall, 60% of the sample reported being employed (625 individuals; not employed: 40%), and only these were contained in the following analyses. Of these, 302 (48.3%) were working from home during the COVID-19 pandemic, and 323 (51.7%) were working on-site as usual. Characteristics of these two groups are outlined in Table 1 and were tested for significant differences with frequency analyses (determining $Chi^2$) and mean differences (MANOVA, reported below).

**Table 1.** Characteristics of individuals working from home (home office, mobile work, etc.) and those working on-site (office, supermarket, hospital, etc.) regarding socio-demographic variables.

| Sample | Worked from Home | Worked on-Site | Total | Bivariate Test Statistic |
|---|---|---|---|---|
| Gender | - | - | - | - |
| Female | 137 (45.4%) | 160 (49.5%) | 297 (47.5%) | - |
| Male | 165 (54.6%) | 163 (50.5%) | 328 (52.5%) | $Chi^2(1) = 1.09; p = 0.30$ |
| Age (years) | M = 38.53 (SD = 13.01) | M = 44.92 (SD = 12.75) | M = 41.61 (SD = 13.26) | $F(1) = 23.27; p < 0.01$ |
| 18–29 years | 82 (27.2%) | 38 (11.8%) | 120 (19.2%) | - |
| 30–39 years | 80 (26.5%) | 60 (18.6%) | 140 (22.4%) | - |
| 40–49 years | 56 (18.5%) | 75 (23.2%) | 131 (21.0%) | - |
| 50–59 years | 53 (17.5%) | 104 (32.2%) | 157 (25.1%) | - |
| 60–69 years | 16 (5.3%) | 37(11.5%) | 53 (8.5%) | - |
| 70+ years | 15 (5.0%) | 9 (2.8%) | 24 (3.8%) | $Chi^2(5) = 47.48; p < 0.01$ |

**Table 2.** Characteristics of individuals working from home (home office, mobile work, etc.) and those working on-site (office, supermarket, hospital, etc.) regarding socio-demographic variables.

| Sample | Worked from Home | Worked on-Site | Total | Bivariate Test Statistic |
|---|---|---|---|---|
| Number of individuals in the household | M = 2.58 (SD = 1.65) | M = 2.40 (SD = 1.20) | M = 2.49 (SD = 1.45) | - |
| 1 | 68 (22.5%) | 77 (23.8%) | 145 (23.2%) | - |
| 2 | 109 (36.1%) | 130 (40.2%) | 239 (38.2%) | - |
| 3+ | 125 (41.4%) | 116 (35.9%) | 241 (38.6%) | $Chi^2(2) = 2.04$; $p = 0.36$ |
| Children | - | - | - | - |
| No | 145 (48.0%) | 188 (58.2%) | 333 (53.3%) | - |
| Yes | 157 (52.0%) | 135 (41.8%) | 292 (46.7%) | $Chi^2(1) = 6.51$; $p = 0.01$ |
| Age of Children * | - | - | - | - |
| Below 1 year | 12 (8.3%) | 11 (5.9%) | 23 (6.9%) | $Chi^2(1) = 0.75$; $p = 0.39$ |
| 1–3 years | 27 (18.6%) | 14 (7.4%) | 41 (12.3%) | $Chi^2(1) = 9.47$; $p < 0.01$ |
| 4–6 years | 27 (18.6%) | 20 (10.6%) | 47 (14.1%) | $Chi^2(1) = 4.30$; $p = 0.04$ |
| 7–10 years | 30 (20.7%) | 26 (13.8%) | 56 (16.8%) | $Chi^2(1) = 2.75$; $p = 0.10$ |
| 11–14 years | 34 (23.4%) | 32 (17.0%) | 66 (19.8%) | $Chi^2(1) = 2.13$; $p = 0.15$ |
| 15–18 years | 63 (43.4%) | 128 (68.1%) | 191 (57.4%) | $Chi^2(1) = 20.32$; $p < 0.01$ |

Note. M = Mean; SD = Standard Deviation; * Age of Children can result from different children.

## 4. Results

Younger employees were more likely to work from home than older employees. Below 40- and above 70-year-old employees made more use of remote work options or were more able to do so because of their job setup. However, there were no significant differences between men and women or regarding the number of people living in the household (see Table 2). Further, it was tested whether having children and the age of the offspring would make a difference.

Analyses revealed that those with children were more likely to work from home than the ones without children. The ones with children in the age group 1 to 6 years were more likely to work from home than to work on-site, which may be related to the need to do so because of closed daycare centers and schools as well as unavailable grandparents and nannies, due to distancing policies to prevent the spread of the COVID-19 virus. Significant differences occurred also with regard to children aged 15–18 years old; those parents were more likely to work on-site, which may also be related to their own age.

To test whether those employees working from home differ from those working on-site with regard to suffering, feeling mentally and/or physically strained from the limitations during the COVID-19 pandemic, and their well-being/health in general, contrasts were computed taking the age effects into account (Figure 2).

A MANCOVA was conducted with age as the covariate and number of individuals living in the household, as well as suffering and mental- and physical-health status, to examine differences due to working from home (factor 1) and gender (factor 2). In this model, only age transpired significant with $F(4, 393) = 7.08$; $p < 0.01$ and $Eta^2 = 0.07$ (see Figure 2), indicating that older individuals were less affected by the COVID-19 pandemic and reported better well-being/mental and physical health. However, after controlling these age differences, working from home or on-site did not significantly interrelate with feeling affected by the COVID-19 pandemic or mental and physical health.

Descriptively, individuals aged 40 (regarding mental aspects) or 50 (regarding physical aspects) seemed to benefit from working from home, whereas the mental health of those aged below 40 appeared better off when maintaining their routine of going to work and not having to work in a home office. However, this was not statistically significant.

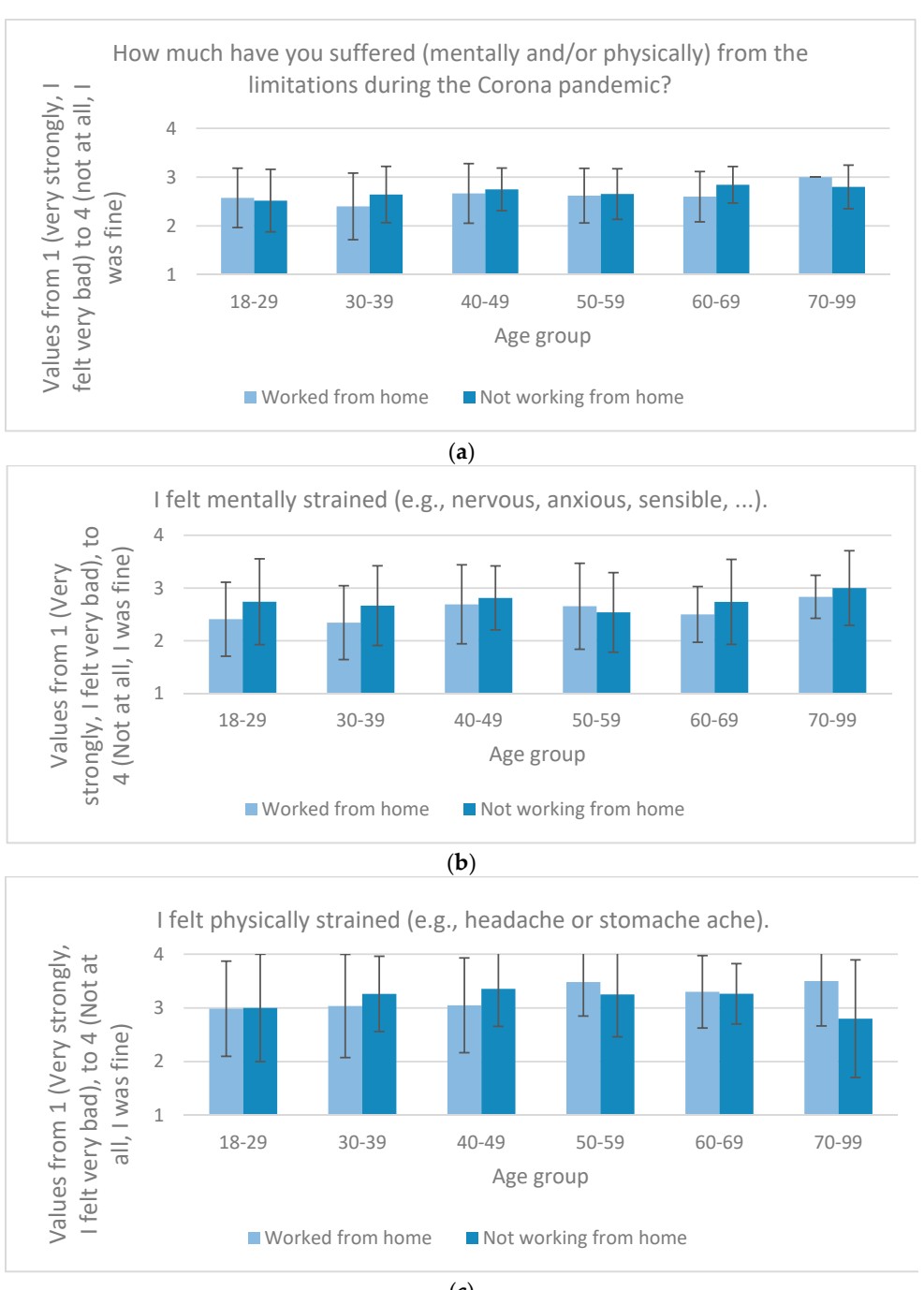

**Figure 2.** (**a**) General Effect of the Corona Pandemic (Mean and Standard deviations), higher means indicate feeling better (less affected). (**b**) Mean rating of mental health status during the Corona pandemic for different age groups working from home vs. not working from home (Mean and Standard deviations), higher means indicate feeling better. (**c**) Mean rating of physical health status during the Corona pandemic for different age groups working from home vs. not working from home (Mean and Standard deviations), higher means indicate feeling better.

Taken together, individuals' perceptions of mobile work were dependent on their age and the situation at home. Those employees having to manage not only working from home but also running other responsibilities such as family duties, seemed to be burdensome. If such responsibilities do not exist, older individuals especially benefit from the option to work from home and make use of digitization and i4.0 in terms of mobile work.

Surely, more aspects should be explicitly taken into account relating to human factors, such as attitudes to change, perceived anxiety, or stress. With this study, we were only able to investigate the aspects that the study participants were interviewed about. Further aspects could also be more positive, such as beneficial consequences of staying home and feeling safer in the face of a pandemic. General factors could be to save the time from commuting for more recreation, more flexibility, and self-determination, which was found to be positively related to employee job satisfaction. As these aspects were not measured in the study, this should be done in the future.

Concluding, employees seem to be ready to make use of mobile work, i.e., digital and also utilizing i4.0 at home. This intrinsic motivation of employees to digitize should now be actively used as a driver of change in order to successfully and sustainably implement digitization in companies [33]. In parallel, we have investigated the general effects of i4.0 technologies on employees and derived solutions for avoiding the associated negative consequences. In the following, these effects are related to aspects (e.g., selection of digital enablers—see Section 2) that need to be taken into account when making decisions in the context of digital transformation in a company. The decision model leads to a software tool that can help companies to reach a good to an optimal decision.

## 5. Discussion

As already predicted by Gattiker and Howg in 1990, digitization/i4.0 technology was leading to major job design change and organizational change [34]. In research, terms similar to "i4.0" in relation to technology are not often associated with the challenge in the aspect of the human component on a personal level or the psychological aspects. These challenges occurred when terms such as "digitization", "decentralization", and "Internet of Things" appeared [1]. Also, Bolander "has pointed to numerous potential weaknesses related to the automation of human decision making" [35] (p. 866). There is a lack of empirical studies or analytical studies with regards to workers' negative responses and behaviors in a digitized environment. Moreover, the symptoms occurring in different sectors, stations, or positions vary significantly. Therefore, the question is how to analyze and describe the cause–effect chain of corporate goals using i4.0 tools, the effects on employees, possible symptoms, and corresponding countermeasures. In order to be able to answer this question, a decision-support model was developed as part of this article (see Figure 3).

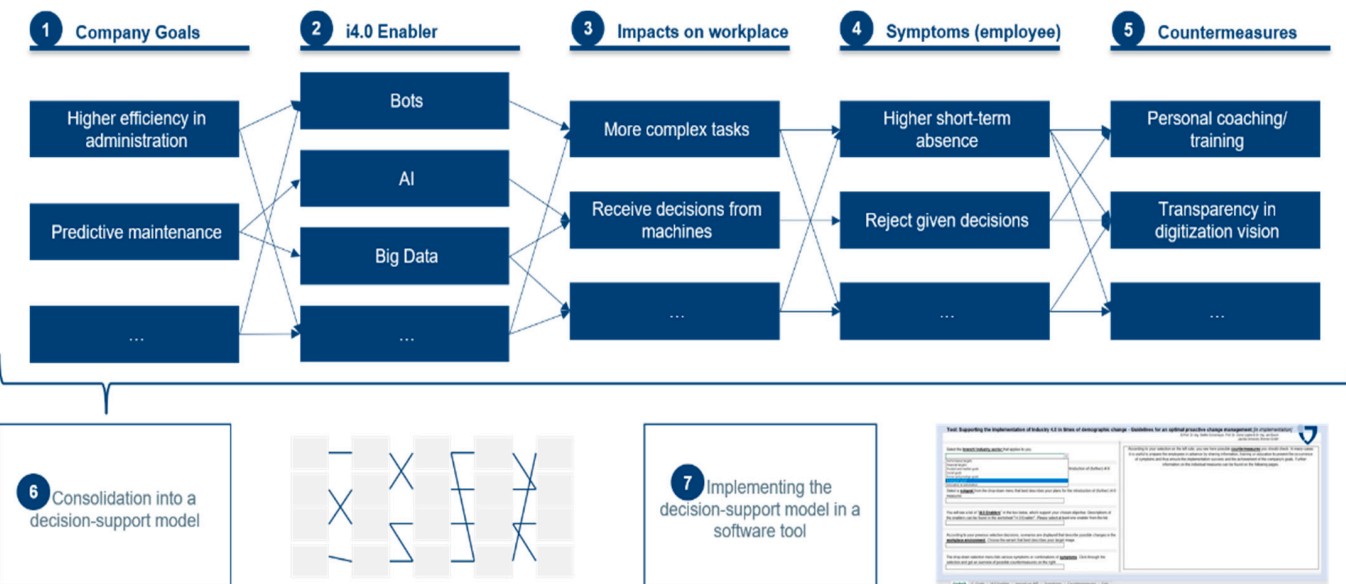

**Figure 3.** Decision-support model.

The seven steps of the procedure were developed accordingly on the basis of the aforementioned preliminary work as well as on our own experience from various industrial digitization projects [36,37]. The seven steps are explained in more detail below and are intended to pave the way for sustainable change.

### 5.1. Company Goals

Companies introduce i4.0 applications, tools, and standards very selectively according to their company goals. The selection of technologies is determined both by the requirements of the industry and, in particular, by the corporate strategy or individual objectives derived from it [33,38]. Individual and relevant technologies are explained in the following Section 5.2, with examples. In order to better understand the motivation and link for selected technologies (see also Section 5.6), typical company goals are explained initially. Moore stated that "All organizations benefit from developing a strategy. The most well-developed strategy models come from the private sector and focus on markets, customers, and competition" [39] (p. 183).

Many previous publications explore new business concepts and strategies for adapting to the new industrial revolution. In this context Crnjac et al. [40] discussed the changes that will occur by applying i4.0. The authors provided an overview of several concepts and strategies and an evaluation of business models linked to i4.0 [40]. Furthermore, they defined eight priority areas for action related to selection and implementing i4.0 [40]. Combining the knowledge and standard approaches of strategy consulting [41] with the topics of sustainability, social responsibility [42], and i4.0, we will divide the overarching company objectives into these seven categories:

- Performance targets: profit and return;
- Financial targets: capital structure and enterprise value;
- Product and market goals: market share, sales revenues, and product range;
- Social goals: jobs, personnel development, job satisfaction, income, and pensions;
- Power and prestige goals: independence and corporate image;
- Ecological goals: conservation of resources and environmental protection;
- Innovation and automation: business bots and robotic process automation.

Accordingly, these aspects should be taken into account when the company goals are translated into the further steps, which will be described in the following.

### 5.2. Industry 4.0/Digital Enablers (with Impacts on Employees)

Certainly, there is a multitude of methods and tools that are counted among the technologies of i4.0. In a huge scientometric survey, scientists found 35 main technologies, which are most frequently discussed in relevant publications. Ranking the technologies based on the number of appearances in these publications in a Pareto chart, 13 key technologies were identified [43]: Internet of Things, Big Data, Additive Manufacturing, Cloud Computing, Autonomous Robots, Virtual Reality and Augmented Reality, Cyber-Physical Systems, Artificial Intelligence and Machine Learning, Smart Sensors, Advanced Simulation, Nanotechnology, Drones, and Biotechnology.

On one hand, the autonomous robots can be industrial robots or robot arms. On the other hand, there are also robots that support employees in carrying out daily work processes. So-called software robots perform tasks mostly similar to humans or by imitating them [44–46]. This process is called Robot Process Automation (RPA). RPA is an approach for automating processes within a wide range of use cases [47]. This method is therefore very well applicable to reaching some of the company goals shown in Section 5.1. For example, some goals include raising profit or achieving higher job satisfaction by letting frequently recurring or boring tasks be performed by RPA.

At this point, the focus should not be on the enumeration and description of i4.0 technologies. It can rather be stated that the toolbox of i4.0 technologies is huge and that each tool or method will have its own impact on the activities of employees. On the one hand, the focus is, of course, on the automation and computerization of workplaces [48–50].

On the other hand, the impact of advanced digital technologies means that some new positions and specializations are associated with new skills and experience [51]. As a result, there will be major changes in the world of employment. For example, using i4.0 technologies, a self-organizing adaptive logistics system can function without significant intervention by employees. Also, the increasing digitization in industrial production can assist employees so that they can assess increasingly complex situations, paying more attention to the key factors and information regarding their tasks (platform Industry 4.0 [52], especially in a time when fast actions are needed, such as during the COVID-19 pandemic). Therefore, the following section is focused on the impact on employees in the course of the introduction of i4.0 technologies.

### 5.3. Industry 4.0 Impact on Workplaces

It is likely that i4.0 technologies have or will soon have huge implications for both work content and work organization and will be able to change the nature of and manner in which employees work [53]. Despite the impact of i4.0 on older, less-educated, and low-paid workers, the continuous progress will result in computers performing tasks that require a high level of education and training. As a result, those employees in highly skilled jobs may feel threatened by machines and software algorithms capable of sophisticated analysis and decision-making. Continued progress in production automation and the deployment of advanced commercial robots would further limit the possibilities for low-skilled workers. Technological progress is relentless, and machines and computers will someday approach the point where they would match or exceed the ability of the average worker to perform most routine tasks [54]. Frey and Osborne carried out a study on the impact of digitization on jobs in the US. The main finding of their study is that 47% of jobs in the US are at risk of being made redundant by computerization. In the future, robots will not only be able to execute standardized programs but will also be able to perform demanding tasks beyond the routine. As a logical outcome, most of the less skilled human jobs would be eliminated and substituted by technology, which would make the surviving human jobs more complex and wide-ranging [49].

Brynjolfsson and McAfee forecast substantial economic changes resulting from the rapidly growing use of i4.0 technologies but are skeptical about the potential positive effects on jobs and expect competition for jobs to increase. In their estimation, technological developments would eradicate not only routine jobs, but also high-skill jobs that are defined by pattern recognition and cognitive non-routine tasks. They also recommend a number of actions to reduce the negative impact of i4.0 technologies and to counterbalance the job losses resulting from the ever-advancing digital technologies [55,56]. On the opposite side, the Boston Consulting Group anticipated a fairly positive future scenario for the impact of i4.0 in an evaluation carried out exclusively for a German management magazine. It put the potential impact of i4.0 technologies at more than 100,000 new jobs in mechanical engineering and construction over a 10-year period. They based their logic on the recognition that the introduction of i4.0 technologies would necessitate a substantial number of additional staff with dedicated technical expertise [53].

Koh and colleagues [53] reviewed the potential impact of i4.0 on corporate, administrative, and management tasks, as well as activities. On the operating work level, they concluded that "Industry 4.0 is transforming jobs and required skills, which have impacts on the working environment and skills development. With more robots and smart machines involved in the daily operation, the physical and virtual world are fusing together, thus launching transformation in the working environment." [53] (p. 823). Like many others, they see a future with greater job enrichment, as new technology would lead to better decision-making and planning decentralization, with the necessity for higher process integrity and cross-functional approaches. Processes such as quality assurance and maintenance would presumably be subject to increased automation, and the growing complexity would most likely call for a greater need for cross-functional management control and real-time capability.

Accordingly, it can be argued that the impact of i4.0 technologies will have different effects. These are in particular the automation of repetitive standard activities, the assumption of more complex tasks by employees and the associated need for employees to acquire new skills and training, and the strong trend towards IT topics and digitization [57]. At this point, it is important to state that the impacts of i4.0 technologies on employees are largely positive. These technologies provide possibilities to make working life more comfortable. One example is already mentioned above in regard to the Robot Process Automation, which allows employees to be relieved from boring standard tasks. Further samples are also listed in the 13 most-significant technologies in Section 5.2. Robots are capable of significantly improving ergonomics at workplaces in assembly lines [58]. In terms of employee development and qualification, simulations, as well as virtual and augmented reality, can increase learning retention, improve problem-solving capabilities, and enable situations with instant feedback in a virtual training environment [59–61].

*5.4. Potential Symptoms of Employees Linked to Industry 4.0*

Job insecurity, negative change attitudes/learning anxiety, and technology anxiety may increase with age. One possibility might also be a feeling of insecurity in terms of losing a job due to i4.0 technologies. Job insecurity can be described as the anxiety of any vertical occupational change. Employees might fear that the technological transformation in the company would disqualify them. A case study concluded that changes in job descriptions could occur in different ways, including transferring tasks from one job description to another, by fusing two or more job descriptions, or by adding new tasks to the old job descriptions [62] (p. 65). Compare also Figure 1: Behavior during change processes in seven discrete phases.

Some possible examples of symptoms in the job insecurity category include disengagement in social events of the company and increased activity on employment-oriented platforms, such as LinkedIn, and higher short-term absence [63–65]. The interviewees in the study of Pfaffinger et al. in 2020 mentioned that information technology experts without experience in leadership could attain a high level of position as the importance of information technology in organizations increases [66]. Increased autonomy occurs when workers are required to learn more leadership skills and self-organizing skills. The perception of the notion "being promoted" generally leads to positive feedback; however, one study also pointed out that increased autonomy or responsibility could also be interpreted as increased accountability [67].

To sum up, employees need to learn new skills while taking into consideration the above-mentioned disadvantages that occur while coping with increased autonomy. This can cause a negative change in attitudes or, in its extreme form, lead to developing a "learning anxiety". More independence could bring about advantages of digitization to make use of a crisis situation such as the COVID-19 pandemic. However, our results, like many other similar findings from times prior to the COVID-19 pandemic, might be interpreted in terms of older people avoiding adapting to digitization because of lifelong learning reservations. This might especially be the case if less learning experience was performed in older age during the past years, because no lifelong learning options or no formal training was provided.

Avoidance learning can be described as a self-protecting response that serves to minimize the perceived threats. In the context of hard-skill acquisition, the employees could develop negative feelings towards the technology. Avoidance, then, could be interpreted as the fear of not being able to master it. Examples for symptoms within the learning anxiety category may include declines in motivation, work quality, and morale in the team, as well as increases in intentional or non-intentional mistakes, absenteeism, and interpersonal conflicts. Furthermore, being uncooperative to formal training and slacking to keep up with the development belong to the potential symptoms [31,66].

Learning anxiety is the main source of resistance to giving up previous beliefs. Learning anxiety can be applied as the anxiety that occurs when one is learning skills to perform

a new task and role [67]. The triggers of learning anxiety are avoidance learning and lack of time for the trainings necessary to keep up while still performing one's current task [66,68].

Technology anxiety refers to anxiety towards digitized technology or technology in general. One study pointed out that there is a high correlation between technology anxiety and technology acceptance [69]. The causes and triggers of the technology anxiety class are an overwhelming amount of information input from the digital devices, lack of knowledge and information, lack of user experience, complexity of the technology user interface, lack of expertise, guidance, and assistance, or even learning anxiety (class of symptoms defined earlier [66]).

A classical review concluded the correlation of "computer anxiety", another closely defined term in the context of technology anxiety, with test and math anxiety [31]. "Computer anxiety inhibits computer learning in a similar way that test anxiety reduces test performance and math anxiety decreases math achievement." [31] (p. 102). This implied that the symptoms in the technology anxiety class could be identified in relevance to work performance in respect to the expected potential. Many studies also mentioned the lack of knowledge of the technology in reference to technology anxiety [31,66,70]. Examples of symptoms from technology anxiety include avoidance of computers or electronics at work or at home, excessive caution with computers or devices given, as well as attempts to cut short the necessary use of these technologies [30,70].

### 5.5. Possible Countermeasures

Based on the defined symptom classes, job insecurity, technology anxiety, and learning anxiety, the corresponding countermeasures are described in the following. One possible countermeasure towards job insecurity in a digitized work environment could be the introduction of prevention and social welfare programs. The latter may step in in the case of job loss in provision of emotional security, to a certain extent, and thereby also work preventatively in terms of diminishing the fear of job loss [66].

For possible countermeasures, a hard distinction between job insecurity and technology anxiety is not appropriate since possible countermeasures mostly help with both types of symptoms. Training or coaching soft skills in the workplace can be considered as a countermeasure for the identified possible symptoms. Exemplary skills are managing financial resources and material resources, individuals' management, time management, decision-making, communication, and leadership skills [71].

Trainings and coaching in the workplace could be very effective as a result of positive organizational outcomes [72,73]. Examples of these outcomes are the performances of the individual, team, and organization. Furthermore, outcomes include skill-based performance, such as leadership skills and top-down/bottom-up management [26], learning competencies, as well as affective performance such as attitude and motivation [73]. Especially in coaching, emphasizing one-on-one stresses assisting the relationship between the coach and the coachee, as well as formally defined agreements in setting goals for personal development, fulfilling the agreement through development, focusing on interpersonal and intrapersonal issues, and using the provision of tools and reinforcements for opportunities the coachee needs for development and growth. The findings of Jones et al. in 2015 showed that coaching would be more effective if training was conducted by internal coaches and when multisource feedback was excluded [73]. Therefore, coaching and training are indispensable in the development of employees, and would directly and indirectly mitigate the symptoms. This is also in line with the finding that employees' participation is needed for continuous improvement of companies' affairs [26].

In an effort to minimize learning anxiety, in 2018, Wirth formulated some solutions that included building visions of the future in presence of the change; formal training on learning competencies; and personal mastery. There were some notable aspects in formulating the training and communication. The management had better focus on the entire group that may be involved. The cultural and peer support was deemed necessary in assistance of the change maintenance. In recommendation of personal mastery, the

learner in the learning process needed to determine the learning objectives, the approach of learning, and learning pace. All of the above targets create sufficient psychological safety for the learner to accept the new information and balance the threats induced from the disconfirmation of the old information [74,75].

*5.6. Consolidation into a Decision-Support Model*

In this part, the systematic structuring is described, which should make the previous steps usable for a model application. Recommendations on what to look for and how to deal with specific symptoms of employees depend on which i4.0 enablers can achieve the corresponding company goals. In order to obtain recommendations for the correct handling of affected employees in digital transformation processes, taking into account the associated enablers and the related objectives, a holistic model is intended to provide support. The model we have developed includes the five procedural steps shown in Figure 3 from the corporate goals to the i4.0 enablers, the impact on employees and their possible symptoms, and suitable countermeasures—and thus provides the relevant recommendations for the executive team. There is no universal solution to guide employees perfectly through the digital transformation but there are certain interactions between these five steps. Depending on which of the goals companies pursue, as described in Section 5.1, different enablers must be selected from the large portfolio of i4.0 technologies. Some enablers make a greater contribution to the achievement of objectives and others correspondingly less. Furthermore, there are connections between the i4.0 enablers and the resulting impacts on the employees concerned. As a function of the effects on their activities and the employees' daily work routines, employees will also show different symptoms. Finally, based on which symptoms the employees show, the appropriate countermeasures must be taken to maintain the employees' efficiency.

If, for example, a company aims to increase the satisfaction of its employees by relieving them of boring and repetitive everyday tasks, one possible i4.0 enabler is Robot Process Automation. The effect of this enabler is that manual, time-consuming, or fault-sensitive activities are automated by software robots. On one hand, the effect that can be expected is that fewer mistakes will be made, which will make the employees more satisfied and give them more time for more interesting tasks. On the other hand, since many standard tasks are no longer necessary, employees suddenly have more time for new tasks. These new tasks may be different from the tasks they have been working on so far or they may even be more demanding or more complex. The effect of this change can lead to feeling overwhelmed and a higher perceived stress level. Possible symptoms that may occur due to this situation include declining motivation, declining work quality, interpersonal conflicts, or higher short-term absenteeism.

Unfortunately, the original intention to increase the satisfaction of the employees through Robot Process Automation, in this case, has quickly transformed into the contrary. If this situation arises, it would be counterproductive to reverse the i4.0 enabler once imposed in order to restore the original situation. Instead, it would be necessary to correctly interpret the symptoms of the employees and to take the appropriate countermeasures. In this case, possible countermeasures can be focused training measures, so that employees do not feel overwhelmed by the new, more complex tasks, or an open discussion with the relevant manager.

In this scenario, the manager can grant the employees the necessary time to familiarize themselves with the new tasks and provide experienced employees with support for a certain period of time. This is just one example of the many possible connections between the five steps shown in Figure 3. By meaningfully combining all identified elements in these five steps, the authors create a holistic model for guiding employees through digital transformation processes. The next section will describe how to convert this model into a software tool.

### 5.7. Implementing the Decision-Support Model in a Software Tool

To support the implementation of i4.0 in times of demographic change, a tool (see Figure 4) is being developed to promote optimal proactive change management. The tool is based on the previously described relationships in the "decision-support model".

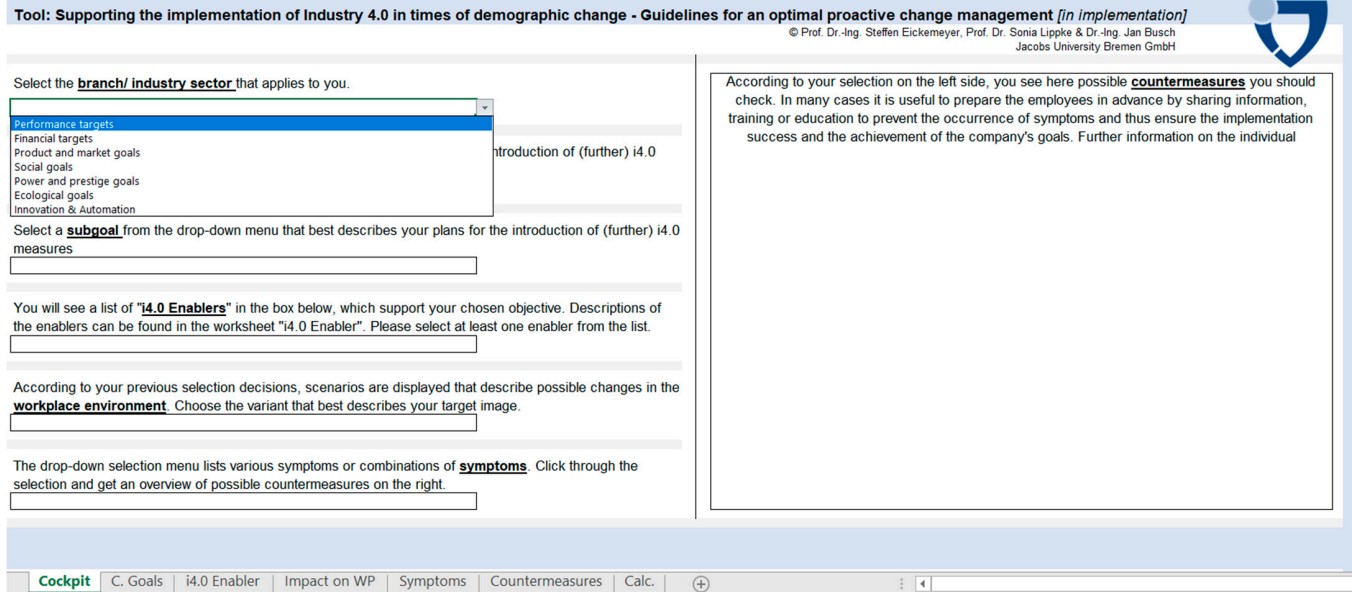

**Figure 4.** Screenshot of the tool's interface.

When using the tool, the user first selects the branch/industry sector that applies to them, and in the second drop-down menu, the target that best describes the individual plans for the introduction of (further) i4.0 measures. This selection concretizes the options for selection in the form of target areas described in more detail. Accordingly, in the next selection field, a sub-target from the drop-down menu must be selected that best describes the plans for the introduction of (further) i4.0 measures.

### 6. Conclusions

The positive impact of digitization on business objectives is well known. However, this manuscript specifically reviewed the challenges involved in the introduction of i4.0 technologies. Currently, most companies are still at the beginning of the implementation of measures for universal i4.0 standards and many companies are not yet as far advanced in this process as they would like to be. Ślusarczyk stated in her study, "it occurs that the level of preparation of enterprises to individual dimensions of i4.0 and the ability to use the benefits of i4.0 are lower than expected, taking into account the positive attitude towards new technologies" [76] (p. 238).

However, there are also many advantages of digitization, especially during a crisis such as the COVID-19 pandemic. Additionally, the ongoing and increasing environmental changes due to pollution and required restrictions in mobility can act as a crisis. Human factors such as negative health effects, well-being, anxiety, or stress need to be taken into account when aiming to help companies and their human capital to adapt accordingly within operations and production management.

The model presented in this manuscript, which leads to a software application, aims to identify the potential negative effects of digitization on employees. By knowing the corresponding causes of negative effects, the executive team can act promptly and correctly and thus efficiently manage the desired digital change. It is not only important to systematically identify the appropriate countermeasures for the affected employees, but it is also essential to eliminate the causes. Thus, the executive management must create a

clear understanding of the overall change process, starting with the goal (see Section 5.1). They must then define the path to this goal in order to be able to determine which changes are required for which departments and employees, and which i4.0 enabler should be implemented. This planned roadmap must be handed over to the managers in such a way that they understand and support the entire process fully.

Only if the managers support the digital transformation will their employees go along with it and help the company to move forward in i4.0. It is always important to take employees' concerns seriously and to respond to them. To support this, the executive management should communicate directly with teams and the individual members that will face the changes.

A participatory approach could help to find sustainable solutions which translate into long-term success, not only for the company but also for the humans working in it, i.e., preventing losing them due to feeling overwhelmed or not wanted/needed anymore. Especially in times of physical distancing, when working from home is not just in the interest of the company to save costs but also needed in terms of public health, to prevent the spreading of a communicable disease such as COVID-19, appropriate solutions are needed. It can be assumed that even after the complete containment of the COVID-19 pandemic, including the mutations, the wheel of time will not be turned back. In all areas of operations and production management, it is clear that mobile work, as a more general form of working from home, will remain an essential component in the future. In this paper, we aimed to outline a possible approach for this situation in order to improve operations and production management. In parallel, other researchers, such as Ślusarczyk, plan to begin research in the same subject area this year: "In the further perspective, the authors suggest expanding the analysis and conducting additional research that will allow for the identification of the links between specific i4.0 technologies as well as the required competences and skills of employees" [77] (p. 13). Due to the activity of several researchers in this field, further research steps can be expected in the coming years by intensifying the discussion.

In summary, the following research limitations and implications can be noted. Although the survey was limited, it does show the importance of employee age. Therefore, industrial application shows that the tool can be used for consulting. Implications for practice are that the understanding of typical developments, such the age of affected employees, can help companies to take appropriate measures. The software tool can thus serve operational and production management to enable efficient digitization. As an implication in the field of social research, it can be stated that employee well-being can be increased by making companies aware of necessary measures, such as training for individuals and groups early in the digitization process. The benefits of the research underlying the paper for readers and for further research are the review, synthesis, and software tool, which provide suggestions for operations and production management practice and stimulate discussion on this for future research. Future research on the optimization of the decision-support model will be conducted according to Ulrich's [78] "strategy of applied research". Since the developed methodology is focused on the challenges of strategic, tactical, as well as operational decisions, for the solution of which practical knowledge is lacking, it belongs to the applied action sciences [78,79]. The better manageability of reality is in the foreground here. Accordingly, the structure of the decision-support model is expanded by us with each digitization project, and opens questions or difficulties encountered by employees on the way to successful implementation of digitization measures, providing the impetus for further subsequent research work.

**Author Contributions:** Conceptualization, S.C.E., C.-T.L. and S.L.; methodology, S.C.E. and S.L.; software, S.C.E.; validation, J.B., S.C.E. and S.L.; formal analysis, S.L.; investigation, S.L.; resources, S.C.E.; data curation, S.L.; writing—original draft preparation, S.C.E.; writing—review and editing, J.B., S.C.E. and S.L.; visualization, S.C.E. and S.L.; supervision, J.B.; project administration, S.L.; funding acquisition, S.L. All authors have read and agreed to the published version of the manuscript.

**Funding:** No funding was received to assist with the preparation of this manuscript. The data collection was funded by the company WELEDA (Project Weleda Trendforschung 2020).

**Institutional Review Board Statement:** Not applicable as all procedures conducted in this study were in accordance with the ethical standards of the 1964 Helsinki declaration and its later amendments or comparable ethical standards.

**Informed Consent Statement:** The individuals provided their written informed consent to participate in this study.

**Data Availability Statement:** The anonymized data is available from the corresponding author.

**Acknowledgments:** We wish to thank Katharina Crefeld, Janine Graumüller and Frau Sarah Melerski for their support with data collection and management. We also thank Tim Thiwissen for selective support from literature research to proofreading. We acknowledge Ronja Bellinghausen for supporting us with manuscript format editing and proofreading.

**Conflicts of Interest:** The authors declare that they have no competing interests.

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
