# Peer review of "Acting Instead of Reacting—Ensuring Employee Retention during Successful Introduction of i4.0"

_asi, doi:10.3390/asi4040097_

Round 1

Reviewer 1 Report

This paper is very interesting and current. The authors conducted a quantitative survey (survey) among a representative sample of 1,050 people aged 18 to determine whether overworking employees could lead to lower labor productivity and how this unfavorable development can be counteracted by measures in the conditions of a major crisis.

Some suggestions in the process of revising this paper:

  • The abstract requires a considerable improvement by highlighting in chronological order the following elements: the research objectives, the methods used, the results obtained and the resulting conclusions;
  • In the abstract, the method must specify the conduct of a survey among the researched population;
  • At the end of the "Introduction" section you must include how the article is organized.
  • I recommend that you enter section 2. Literature review before "1.1. Industry 4.0" and expand this section;
  • Section 2. Materials and Methods need to be improved.
  • When conducting a quantitative (marketing) research we must highlight the following aspects:

(1) Determining the type of survey and the advantages of its choice in accordance with the purpose of the research and the way of communication with the respondents.

(2) Sample size and method of determination (for example: probabilistic sampling of the stratified sampling type)

(3) The way of designing the questionnaire in order to generate the necessary data to achieve the research objectives, the validation of the questionnaire, etc.

  • I recommend entering the academic and practical contribution;
  • What is the future of research?

Author Response

Review 1

This paper is very interesting and current. The authors conducted a quantitative survey (survey) among a representative sample of 1,050 people aged 18 to determine whether overworking employees could lead to lower labor productivity and how this unfavorable development can be counteracted by measures in the conditions of a major crisis.

RESPONSE BY THE AUTHORS: Thank you very much for appreciating our manuscript and the work with it!

Some suggestions in the process of revising this paper:

  • The abstract requires a considerable improvement by highlighting in chronological order the following elements: the research objectives, the methods used, the results obtained and the resulting conclusions;

RESPONSE BY THE AUTHORS: Thank you for this feedback, we have accordingly revised the abstract by means of ensuring that we follow this structure (pls see below in bracket):

(Background) The increasing implementation of digital technologies has various positive impacts on companies. However, many companies often rush into such an implementation of technological trends without sufficient preparation and pay insufficient attention to the human factors involved in digitization. This phenomenon can be exacerbated when these technologies become highly dependent, as during the Corona pandemic. (Research objectives) This study aims to better understand challenges and to propose solutions for a successful implementation of digitized technology. (Methods) A literature review is combined with survey results and specific consulting strategies. Data from the first wave of the COVID-19 pandemic in Germany was collected by means of an online survey with a representative sample of the German population. (Results) We did not reveal any correlation between home office and suffering, mental health, and physical health (indicators of digitization usage to cope with COVID-19 pandemic), but rather that younger workers are more prone to using digitized technology. Based on previous findings that older individuals tend to have negative attitudes toward digital transformation, appropriate countermeasures are needed to help them become more tech-savvy. Accordingly, a software tool is proposed. (Conclusions) The tool can help the management team to manage digitization efficiently. Employee well-being can be increased as companies are made aware of necessary measures such as training for individuals and groups at an early stage.

However, we did not include that into the manuscript because we did not want to disobey https://www.mdpi.com/journal/asi/instructions (Abstract: The abstract should be a total of about 200 words maximum. The abstract should be a single paragraph and should follow the style of structured abstracts, but without headings: 1) Background: Place the question addressed in a broad context and highlight the purpose of the study; 2) Methods: Describe briefly the main methods or treatments applied. Include any relevant preregistration numbers, and species and strains of any animals used. 3) Results: Summarize the article's main findings; and 4) Conclusion: Indicate the main conclusions or interpretations. The abstract should be an objective representation of the article: it must not contain results which are not presented and substantiated in the main text and should not exaggerate the main conclusions.)

  • In the abstract, the method must specify the conduct of a survey among the researched population;

RESPONSE BY THE AUTHORS: Thank you for this suggestion, we have accordingly revised the abstract by means of including the sentence now: “Data from the first wave of the COVID-19 pandemic in Germany was collected by means of an online survey with a representative sample of the German population.“ (line 17ff).

  • At the end of the "Introduction" section you must include how the article is organized.

RESPONSE BY THE AUTHORS: Thank you very much for this feedback. Accordingly, we have integrated an overview of the following chapters at the end of the Introduction to describe the structure of the article to the reader:

“The article is structured in 6 chapters: the following chapter 2 presents a literature review to classify the article thematically. Chapter 3 describes the empirical evidence with the materials and methods used. Based on this, the results of the survey are presented in chapter 4. The findings and the relationships described are discussed in chapter 5 and used to develop a decision support model for dealing with employees during digital transformation. Finally, chapter 6 contains the conclusions, which pro-vide both a summary of the article and an outlook on subsequent research work.” (lines 78ff).

  • I recommend that you enter section 2. Literature review before "1.1. Industry 4.0" and expand this section;

RESPONSE BY THE AUTHORS: Thank you for this feedback. We have accordingly inserted chapter 2 Literature review before the chapter Industry 4.0 and explained how the three super chapters of the literature review build on each other to enable the reader to optimally understand the article:

“The following is a summary of the areas of literature that are essential to this paper. First, an understanding of digitization (chapter 2.1), which is central to this article, is created. In the context of this, the central buzzword Industry 4.0, which is the driver for many industry initiatives, is brought into focus. Chapter 2.2 provides a general ex-planation of human behavior during change processes. These two chapters help to provide a basic understanding, so that Chapter 2.3 can then focus specifically on ef-fects on employees during implementation of Industry 4.0. These effects on employees and the correct way to deal with them are the main topic for the rest of this article.” (lines 86ff).

  • Section 2. Materials and Methods need to be improved.

RESPONSE BY THE AUTHORS: Thank you for this feedback. We have accordingly revised the section “2. Materials and Methods”.

  • When conducting a quantitative (marketing) research we must highlight the following aspects:

(1) Determining the type of survey and the advantages of its choice in accordance with the purpose of the research and the way of communication with the respondents.

RESPONSE BY THE AUTHORS: Thank you for the opportunity to expand on this aspects. We accordingly added the following aspects i.e. sentences in red: “The survey by means of online questionnaire was launched in June 2020 to collect data from a sample with a sample size of N=1,050 individuals aged 18 and above by Bilendi GmbH on behalf of Weleda. A representative sample of inhabitants in Germany was recruited. The questionnaire-based survey was conducted between June 08 and June 15 in the year 2020. Using email, the company contacted individuals in their database by means of stratified sampling. In the inviting email, the purpose of the study was explained. When individuals clicked on the link to the study, they received further participant information which included a confidentiality statement. After reading this, study participants were asked to give informed consent by filling in a form. Only those who provided informed consent were allowed to continue to the questionnaire. The reason for this methodology and advantages of its choice were that the research questions could be investigated at high quality and easy access to the sample even in times of restrictions due to the COVID-19 pandemic. The main aim was to be able to generalize the findings to the larger population and to overcome bias in typical respondent behaviors.“ (lines 195ff).

(2) Sample size and method of determination (for example: probabilistic sampling of the stratified sampling type)

RESPONSE BY THE AUTHORS: Thanks. We included this information now explicitly: “…sample size of N=1,050…” (line 196) and “stratified sampling“ (line 200).

(3) The way of designing the questionnaire in order to generate the necessary data to achieve the research objectives, the validation of the questionnaire, etc.

RESPONSE BY THE AUTHORS: We appreciate the option to give more details. Accordingly, we now report: “

The way of designing the questionnaire in order to generate the necessary data to achieve the research objectives was determined by a theory-based and co-creative process: The last author performed a scoping review of the literature and in exchange with potential future study participants, items and response formats were piloted and partially adapted. The validation of the questionnaire was performed by means of face validity and pilot analyses due to limited options to cross-validate the measures with other items.“ (lines 211ff).

  • I recommend entering the academic and practical contribution;

RESPONSE BY THE AUTHORS: Thank you for this feedback. We have accordingly added the academic and practical implications, and the section now reads:

“Implications for practice are that the understanding of typical developments like the age of affected employees can help to take appropriate measures. The software tool can thus serve operational and production management to enable efficient digitization. As an implication in the field of social research, it can be stated that employee well-being can be increased by making companies aware of necessary measures such as training for individuals and groups early in the digitization process. The benefits of the research underlying the paper for readers and for further research are the review, synthesis, and software tool, which provide suggestions for operations and production management practice and stimulate discussion on this for future research.” (lines 649ff)

  • What is the future of research?

RESPONSE BY THE AUTHORS: Thank you for this feedback. We have accordingly added the future of research, and the section now reads:

“Future research on the optimization of the decision support model will be conducted according to Ulrich's [78] "strategy of applied research". Since the developed method-ology is focused on the challenges of strategic, tactical as well as operational decisions, for the solution of which practical knowledge is lacking, it belongs to the applied ac-tion sciences [78, 79]. The better manageability of reality is in the foreground here. Ac-cordingly, the structure of the decision support model is expanded by us with each digitization project, and open questions or difficulties encountered by employees on the way to successful implementation of digitization measures provide the impetus for further subsequent research work.” (lines 658ff).

Reviewer 2 Report

Congratulations to the authors of this article, which is ambitious and important in the current reality. It was written to the highest standards. The article will be important for academics dealing with these issues as well as it will have a utilitarian dimension - i.e. it will have a very large contribution to the practice of company and employee management.
The goals formulated in the article are very accurate, and the methods selected are appropriate to conduct the planned research. Everything is preceded by an extensive and accurate literature review. The discussion of the results was also very insightful. Interesting Decision-support model which is an important contribution of the authors in the conducted research.
The only remark I would like to add is the request to indicate the limitations accompanying the conducted research.
I also recommend (if the authors find it appropriate) to read the following publications:

Ĺšlusarczyk, B. (2018). Industry 4.0: Are we ready?. Polish Journal of Management Studies, Vol.17, No.1, pp. 232-248

Afonasova, M. A., Panfilova, E. E., Galichkina, M. A., & Ĺšlusarczyk, B. (2019). Digitalization in economy and innovation: The effect on social and economic processes. Polish Journal of Management Studies, Vol. 19, No.2, pp. 22-32

Ĺšlusarczyk, B., TvaronaviÄŤienÄ—, M., Haque, A. U., & Judit, O. L. Á. H. (2020). Predictors of Industry 4.0 technologies affecting logistic enterprises’ performance: International perspective from economic lens. Technological and economic development of economy26(6), 1263-1283.

Ĺšlusarczyk, B., Nathan, R. J., & PypĹ‚acz, P. (2021). Employee Preparedness for industry 4.0 in logistic sector: A cross-national study between Poland and Malaysia. Social Sciences10(7), 258.

Author Response

Review 2

Congratulations to the authors of this article, which is ambitious and important in the current reality. It was written to the highest standards. The article will be important for academics dealing with these issues as well as it will have a utilitarian dimension - i.e. it will have a very large contribution to the practice of company and employee management.
The goals formulated in the article are very accurate, and the methods selected are appropriate to conduct the planned research. Everything is preceded by an extensive and accurate literature review. The discussion of the results was also very insightful. Interesting Decision-support model which is an important contribution of the authors in the conducted research.

RESPONSE BY THE AUTHORS: Thank you very much for appreciating our work.

The only remark I would like to add is the request to indicate the limitations accompanying the conducted research.
I also recommend (if the authors find it appropriate) to read the following publications:

Ĺšlusarczyk, B. (2018). Industry 4.0: Are we ready?. Polish Journal of Management Studies, Vol.17, No.1, pp. 232-248

Afonasova, M. A., Panfilova, E. E., Galichkina, M. A., & Ĺšlusarczyk, B. (2019). Digitalization in economy and innovation: The effect on social and economic processes. Polish Journal of Management Studies, Vol. 19, No.2, pp. 22-32

Ĺšlusarczyk, B., TvaronaviÄŤienÄ—, M., Haque, A. U., & Judit, O. L. Á. H. (2020). Predictors of Industry 4.0 technologies affecting logistic enterprises’ performance: International perspective from economic lens. Technological and economic development of economy26(6), 1263-1283.

Ĺšlusarczyk, B., Nathan, R. J., & PypĹ‚acz, P. (2021). Employee Preparedness for industry 4.0 in logistic sector: A cross-national study between Poland and Malaysia. Social Sciences10(7), 258.

RESPONSE BY THE AUTHORS: Thanks for these constructive suggestions. Accordingly, we have read the publications mentioned and included two papers. In connection with this, we have also added the limitation mentioned, and the sections now are:

(1) Currently, most companies are still at the beginning of the implementation of measures for universal i4.0 standards and many companies are not yet as far advanced in this process as they would like to be. Ĺšlusarczyk stated in her study, “it occurs that the level of preparation of enterprises to individual dimensions of i4.0 and the ability to use the benefits of i4.0 are lower than expected, taking into account the positive attitude towards new technologies” [76] (p. 238). (lines 601ff).

(2) In parallel, other researchers such as Ĺšlusarczyk plan to begin research in the same subject area this year: “In the further perspective, the authors suggest expanding the analysis and conducting additional research that will allow for the identification of the links between specific i4.0 technologies as well as the required competences and skills of employees” [77] (p. 13). Due to the activity of several researchers in this field, fur-ther research steps can be expected in the coming years by intensifying the discussion. (lines 640ff).

(3) In summary, the following research limitations and implications can be noted: Although the survey was limited, it does show the importance of employee age. Thereby, industrial application shows that the tool can be used for consulting. (lines 647ff).
